# Convenient iron-catalyzed reductive aminations without hydrogen for selective synthesis of N-methylamines

Kishore Natte[1], Helfried Neumann[1], Rajenahally V. Jagadeesh[1] & Matthias Beller [1]

N-Methylated amines play an important role in regulating the biological and pharmaceutical properties of all kinds of life science molecules. In general, this class of compounds is synthesized via reductive amination reactions using high pressure of molecular hydrogen. Thus, on laboratory scale especially in drug discovery, activated (toxic) methyl compounds such as methyl iodide and dimethyl sulfate are still employed, which also generate significant amounts of waste. Therefore, the development of more convenient and operationally simple processes for the synthesis of advanced N-methylamines is highly desired. Herein, we report the synthesis of functionalized and structurally diverse N-methylamines directly from nitroarenes and paraformaldehyde, in which the latter acts as both methylation and reducing agent in the presence of reusable iron oxide catalyst. The general applicability of this protocol is demonstrated by the synthesis of >50 important N-methylamines including highly selective reductive N-methylations of life science molecules and actual drugs, namely hordenine, venlafaxine, imipramine and amitriptyline.

---

[1] Leibniz-Institute for Catalysis at the University of Rostock, Albert-Einstein-Straße 29 a, D-18059 Rostock, Germany. Correspondence and requests for materials should be addressed to R.V.J. (email: Jagadeesh.Rajenahally@catalysis.de) or to M.B. (email: Matthias.Beller@catalysis.de)

The development of new and convenient methodologies for the synthesis and modification of advanced life science intermediates remains an important goal for chemistry. Most of the known agrochemicals and pharmaceuticals contain amino groups, which represent a key scaffold in the vast majority of bio-active compounds. Hence, the synthesis and functionalization of amines continues to attract the interest of researchers from chemistry, biology, and medicine. Among the known reactions of amines, especially N-methylation is of importance to regulate the biological and pharmaceutical activities of life science molecules[1–6]. For example, in nature N-methylation of peptides and DNA controls biological functions and plays a vital role in epigenetic changes in gene expression for cellular phenotypes[6]. Interestingly, this comparatively small structural change can activate large protein complexes and also controls the action of enzymes and antibodies as well as the pharmacokinetics and drug delivery[1,2]. In addition, N-methylation of small bioactive molecules modulates the cytotoxicity and importantly makes them more lipophilic which enhances their solubility in bio-membranes[7]. Consequently, top selling drugs such as olanzapine, oxycodone, imatinib, viagra and venlafaxine contain N-methylamino groups (Fig. 1), which play a significant role in their activities[8].

Besides these biological functions, N-methylamines represent important intermediates for bulk and fine chemicals as well as materials[9–11]. More specifically, methylamines such as $MeNH_2$, $Me_2NH$ and $Me_3N$ are produced in >1 million tons per year by reaction of ammonia with methanol[12]. Notably, there is an increasing demand on these products, which grow annually by 4–5%. Unfortunately, the applied catalysts require drastic conditions (>200 °C) and do not allow to produce higher value N-methylamines. Hence, these products are often synthesized by reductive N-methylation reactions. Among them, the Eschweiler–Clarke methodology[13,14] and reductive aminations, in which the corresponding amines and formaldehyde are converted to N-methylamines in the presence of hydrogen or stoichiometric reducing agents[15–17], prevail.

To perform selective reductive aminations of more advanced and multi-functionalized substrates the use of a proper catalyst is crucial. So far most of them are based on noble metals[18]. However, the development of earth-abundant metal catalysts is becoming increasingly important. In this regard, in the past decade especially iron became highly attractive due to its abundance (4.7% in the earth's crust; second most abundant metal), bio-relevance and low toxicity, which makes it not only an ideal metal for catalysis but also for drug discovery and synthesis[19–32]. For example, molecular-defined iron complexes have been shown to promote catalytic hydrogenations[27–29], dehydrogenations[30,31], and aminations[26,32]. Despite these elegant achievements, heterogeneous iron catalysts are preferable due to their stability, reusability and easy separation.

In addition, it should be noted that a general problem of all these hydrogenations is the necessity of special equipment and the need for additional pressure of hydrogen. Therefore, on laboratory scale and in drug discovery still the use of activated but toxic methyl compounds is popular[33,34]. Although in recent years, more benign N-methylations using methanol[35,36] and $CO_2$[37–39] have been disclosed, these methods are restricted regarding sensitive functional groups and substrate scope. Thus, the use of convenient reagents for selective methylation of life science molecules continues to be an important goal. In this regard, paraformaldehyde[40–42], which is stable and easy to handle, can be a suitable methylation reagent.

Here, we show that reductive aminations of nitroarenes with paraformaldehyde proceed in the presence of an earth-abundant and reusable iron oxide-based nanocatalyst. The developed straightforward, convenient and step economic process avoids the necessity of any specialized equipment and also the need of additional reducing agents. Applying this operationally simple protocol, we synthesized a broad series of functionalized and structurally diverse N-methylamines. The synthetic utility of this methodology is specifically demonstrated using various life science molecules including existing pharmaceuticals.

## Results

**Reaction Design**. In general, N-methylated (hetero)aromatic amines are prepared in a two-step sequence from easily available nitroarenes (Fig. 2). Obviously, a straightforward direct transformation of nitroarenes is advantageous regarding step economy and price of substrates[43–46]. So far, noble metal-based catalysts, e.g., Pd/C, prevail in this reaction. To compare the activity of our recently developed $Fe_2O_3$/NGr@C material[19, 44] with the common Pd/C catalyst, the reductive N-methylation of 4-nitroanisole **1** with paraformaldehyde was used as a benchmark reaction. As shown in Table 1, both catalysts exhibited similar activity at different hydrogen pressure (1–50 bar) and produced the corresponding di-methylated amine **2** in 76–87% yield, respectively (Table 1, entries 1–6). Surprisingly, even without any external hydrogen this dimethylation reaction proceeded. However, it is important to note that under the latter conditions the $Fe_2O_3$/NGr@C-catalyst is significantly more active compared to Pd/C (Table 1, entries 7 and 8). As expected in the presence of the parent homogeneous iron complex or commercial $Fe_2O_3$, as well as pyrolyzed iron acetate on carbon no activity was observed (Supplementary Table 1, entries 1–8). However, related iron nanoparticles prepared by the pyrolysis of iron acetate-amine complexes, e.g., 2,2′-bipyridine, 2,2′;6′,2″-terpyridine, and 2,6-bis (2-benzimidazolyl)pyridine at varying temperatures on different heterogeneous supports showed some activity (Supplementary Table 1, entries 9–18).

**Synthesis of N-Methyl- and N,N-dimethylamines**. After having a convenient protocol in hand for the model reaction, we investigated the reductive N-methylation of a broad range of

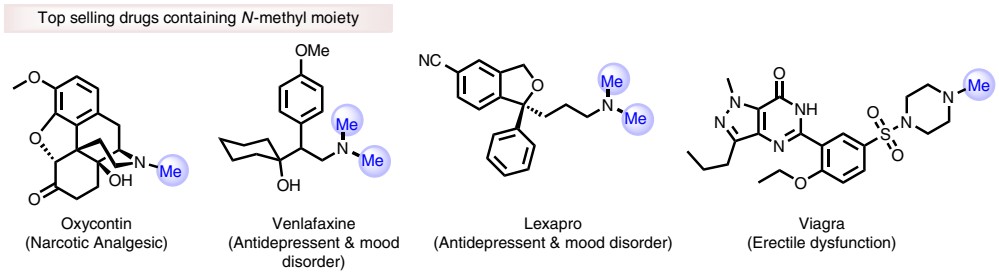

**Fig. 1** Top selling drugs: Selected important drugs containing N-methyl moiety

substrates. In general, industrial bulk nitroarenes but also more demanding functionalized and heteroaromatic nitro compounds gave the corresponding *N,N*-dimethylamines in good to excellent yields (Fig. 3a–d). For organic synthesis it is of special importance to achieve chemoselective reductive *N*-methylation of nitroarenes in the presence of sensitive functional groups such as halogen, alcohols, thiols, ketone, ester, amide, sulfonamide and C–C triple and double bonds (Fig. 3b, c). Gratifyingly, in presence of all these functionalities the nitro group is selectively transformed to the corresponding *N,N*-dimethylamine. Heterocyclic diamines, which are important substructures in active life-science molecules, agrochemicals and advanced materials, are obtained up to 94% using this reductive amination protocol. As an example, 4-dimethylaminopyridine (DMAP), a useful nucleophilic catalyst for a variety of reactions is prepared in a straightforward manner.

To demonstrate the applicability of this methodology further on, the methylation of nitro-substituted biologically active molecules to the corresponding *N, N*-dimethylated analogs was investigated (Fig. 4). Notably, reaction of calcium channel blockers (CCBs)[47] which represent important drugs such as nimodipine, clinidipine and nicardipine produced the desired *N,N*-dimethylamines in up to 76% yield (Fig. 4). In all cases, the

nitro group is highly selective transformed without affecting the core structure of the CCBs. In addition, dimethylation of nimesulide, a non-steroidal anti-inflammatory drug (NSAID) with analgesic and antipyretic properties[48], proceeded smoothly in 79% yield. Furthermore, rhodamine and fluorenone derivatives, which are widely used as fluorescent probes[49] were successfully transformed (Fig. 4).

from *N, N*-dimethylation also selective mono-methylation is possible with paraformaldehyde. Initially, we demonstrated this applying rhodamine derivative 49 (Fig. 5a). Here, the selectivity is easily controlled by the concentration of paraformaldehyde and reaction time. In addition to 51, five selected mono-methylated anilines were synthesized in good yields (Fig. 5b)

**Methylation of aliphatic amines and application to pharmaceuticals.** Apart from nitroarenes, our protocol allows for the use of a variety of bio-active primary and secondary amines which can be exploited for drug discovery. Hence, this benign and convenient reductive amination process is amenable for late-stage synthetic manipulation of all kinds of *N*-based life science molecules. In this respect, it is important to note that

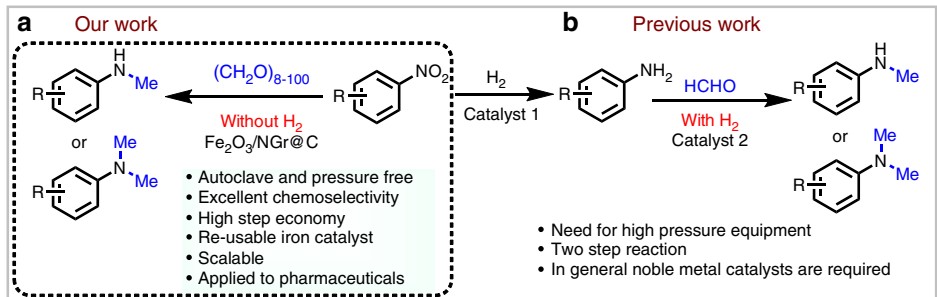

**Fig. 2** Reductive methylation of nitroarenes to *N*-methylamines. **a** Present work representing the straight forward synthesis of *N*-methylamines from nitroarenes without external hydrogen. **b** Previous works showing two-step process for the preparation of *N*-methylamines in presence of external hydrogen

**Table 1 Reductive *N*-methylation of 4-nitroanisole using Pd/C- and Fe₂O₃/NGr@C—catalysts[a]**

| Entry | Catalyst | H₂ | Yield of 2 (%) |
|---|---|---|---|
| 1 | Pd/C | 50 bar | 77 |
| 2 | Fe₂O₃/NGr@C | 50 bar | 87 |
| 3 | Pd/C | 5 bar | 76 |
| 4 | Fe₂O₃/NGr@C | 5 bar | 86 |
| 5 | Pd/C | 1 bar | 80 |
| 6 | Fe₂O₃/NGr@C | 1 bar | 85 |
| 7 | Pd/C | – | 30 |
| 8 | Fe₂O₃/NGr@C | – | 86 |

Pd/C = Commercial catalyst with 5 wt% Pd. Fe₂O₃/NGr@C = pyrolyzed Fe-phenanthroline complex on carbon at 800 °C for 2 h under argon with 2.95 wt% Fe. Reaction conditions: 0.5 mmol 4-nitroanisole, weight of catalyst corresponds to 5 mol% metal (53 mg Pd/C; 50 mg Fe₂O₃/NGr@C), 10 mmol paraformaldehyde (300 mg). 1 mmol Na₂CO₃ (106 mg), 2 mL DMSO-water (1:1), 130 °C, 30 h. Yields are determined by GC using n-hexadecane as internal standard

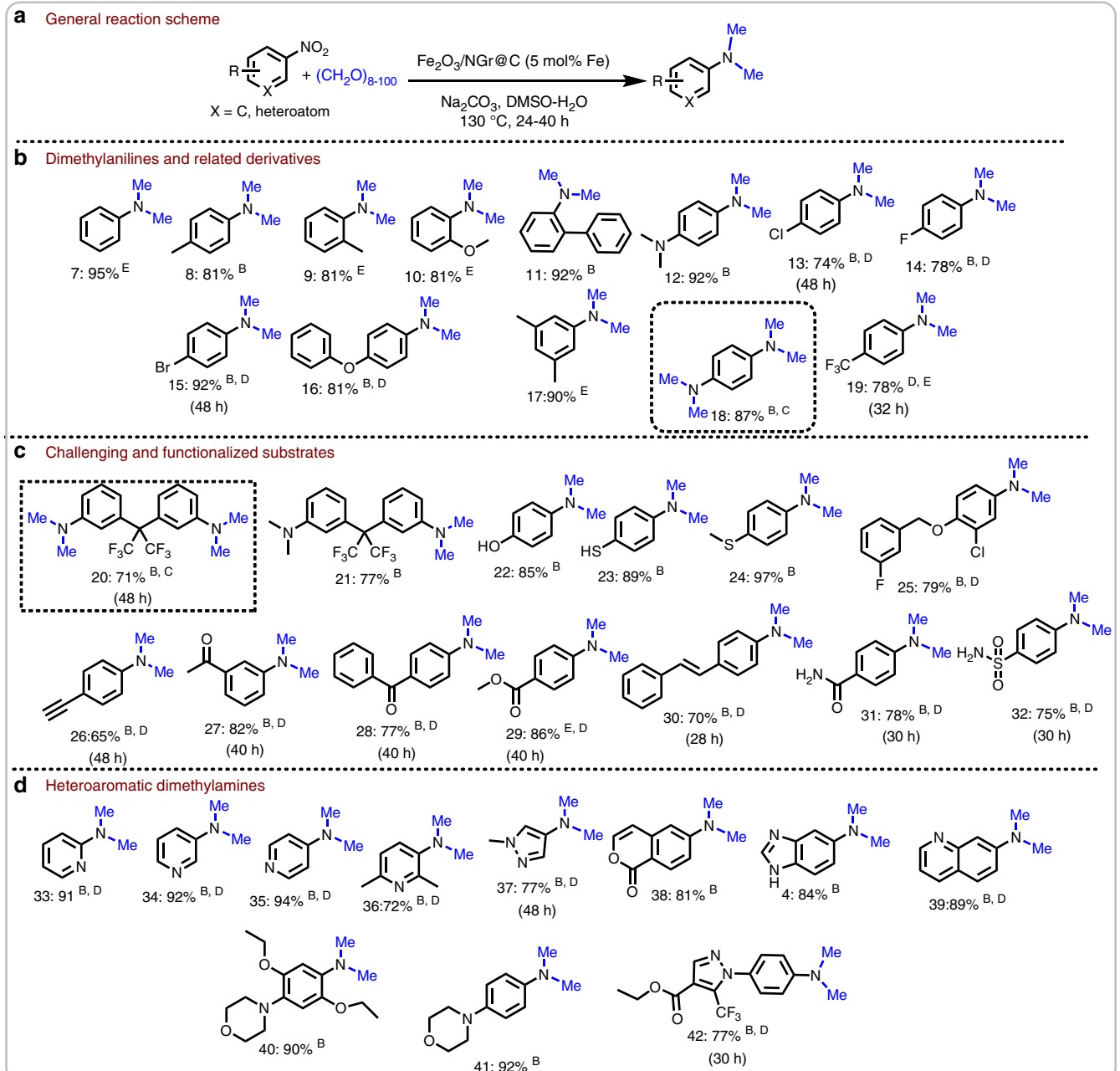

**Fig. 3** Synthesis of (hetero)aromatic N,N-dimethylamines[A]. **a** General scheme representing reductive amino-methylation. **b** Synthesis of dimethylanilines and related derivatives. **c** Synthesis challenging and functionalized dimethylamines. **d** Synthesis of heteroaromatic dimethylamines [A]Reaction conditions: 0.5 mmol nitroarene, 50 mg Fe₂O₃/NGr@C, (5 mol% Fe), 10 mmol of paraformaldehyde (300 mg), 1 mmol Na₂CO₃, 2 mL DMSO-water (1:1), 130 °C, 24 h. [B]Isolated yield. [C]100 mg of Fe₂O₃/NGr@C, 20 mmol (600 mg) paraformaldehyde, 2 mmol of Na₂CO₃, 4 mL DMSO-water (1:1), 130 °C, 48 h. [D]50 mg Fe₂O₃/NGr@C, 20 mmol of paraformaldehyde (600 mg), 1.5 mmol Na₂CO₃, 130 °C, 24 h. [E]Yields were determined by GC

governmental regulations impede the use of toxic metal catalysts at this stage. In this respect, our heterogeneous iron catalyst provides a solution due to its low toxicity and easy removal. Indeed, the methylation of functionalized amines is demonstrated by the synthesis of important existing pharmaceuticals which belong to the 200 top selling drugs (Fig. 6). For example, the naturally occurring alkaloid hordenine (N, N-dimethyltyramine)[50] is prepared in excellent yield from its amine intermediate (Fig. 6). Further, venlafaxine (Effexor), which is used for the treatment of depression and anxiety disorders, is obtained in 89% yield (Fig. 6)[51]. In addition, reductive methylation of mono-

methylated desipramine and nortriptyline gave imipramine and amitriptyline, respectively (Fig. 6)[52]. Moreover, methylation of important active life science molecules such as amlodipine, cinacalcet, duloxetine and sertraline proceeded smoothly (92–95% yields; Fig. 6) without affecting other functionalities or the core-structure of the molecules.

**Gram-scale reactions and recycling of the catalyst**. Finally, we wanted to demonstrate the practical utility of our protocol. For this purpose five different N,N-dimethylamines were successfully synthesized on 1–10 g scale (Fig. 7).

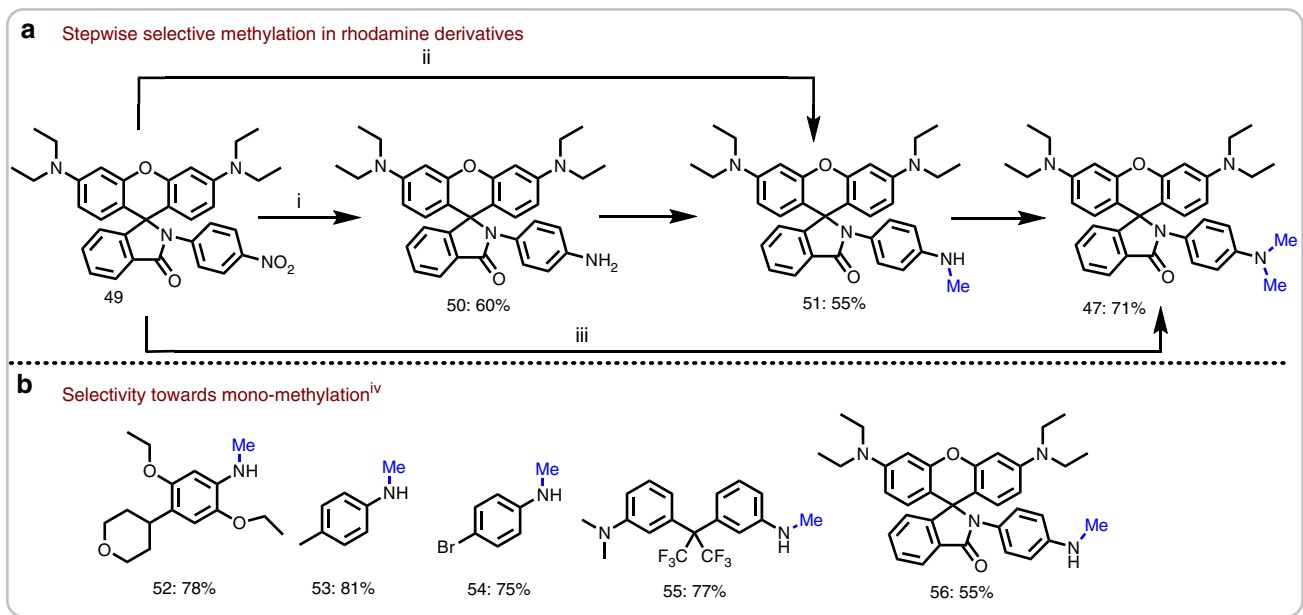

**Fig. 4** N-methylation of pharmaceutical and fluorescent molecules[a]. Selective conversion of nitro groups to N-methyl moiety in selected drug and florescent molecules using iron catalyst. Iron-catalyzed reductive amino-methylation of existing drug molecules [a]Reaction conditions: 0.5 mmol nitroarene, 50 mg Fe$_2$O$_3$/NGr@C, (5 mol% Fe), 10 mmol paraformaldehyde (300 mg), 1 mmol Na$_2$CO$_3$, 2 mL DMSO-water (1:1), 130 °C, 30 h, isolated yields. [b]same as 'a' with 20 mmol (600 mg) paraformaldehyde

**Fig. 5** Controlled stepwise reductive methylation. **a** Example showing fluorescent compound[i, ii, iii]. **b** Selectivity towards mono-methylation[iv]. [a]Reaction conditions **a**: [i]0.5 mmol of 49, 50 mg Fe$_2$O$_3$/NGr@C (5 mol% Fe), 2 mmol paraformaldehyde, 1 mmol Na$_2$CO$_3$, 2 mL DMSO-water (1:1), 130 °C, 8 h. (**b**): [ii]0.5 mmol of 49, 50 mg Fe$_2$O$_3$/NGr@C (5 mol% Fe), 3.2 mmol paraformaldehyde, 1 mmol Na$_2$CO$_3$, 2 mL DMSO-water (1:1), 130 °C, 18 h. [iii]0.5 mmol of 49, Fe$_2$O$_3$/NGr@C (5 mol% Fe), 10 mmol paraformaldehyde, 1 mmol Na$_2$CO$_3$, 2 mL DMSO-water (1:1), 130 °C, 30 h. [iv]Reaction conditions of (B): 0.5 mmol nitroarene, 50 mg Fe$_2$O$_3$/NGr@C (5 mol% Fe), 3.2 mmol paraformaldehyde, 1 mmol Na$_2$CO$_3$, 2 mL DMSO-water (1:1), 130 °C, 18 h. All are isolated yields

In addition, we demonstrated the stability and re-use of the catalyst. Obviously, such recycling is important for the advancement of cost-effective process development. Notably, our iron catalyst is stable and conveniently recycled up to 5 times without any significant loss of catalytic activity (Fig. 8).

## Discussion

To understand the reasons for this superior activity of the iron catalyst, the mechanism of the underlying reduction process was investigated. Hence, paraformaldehyde was reacted in the presence of Fe$_2$O$_3$/NGr@C at 130 °C for 4 h. Gas analysis (GC) revealed the formation of H$_2$, CO and CO$_2$. This result clearly indicates that paraformaldehyde is converted under comparably mild conditions to syngas (CO + H$_2$). Subsequent water-gas shift reaction of CO in the presence of water produced additional H$_2$ and CO$_2$. Evidently, the small amounts of in situ generated hydrogen allow for selective reduction of nitroarene and the corresponding imine after condensation with formaldehyde to yield the mono-N-methylated product. Following a second condensation with formaldehyde followed by reduction again finally gives the N,N-dimethylamine (Fig. 9a). Control experiments using [13]C-labeled paraformaldehyde showed that the methyl

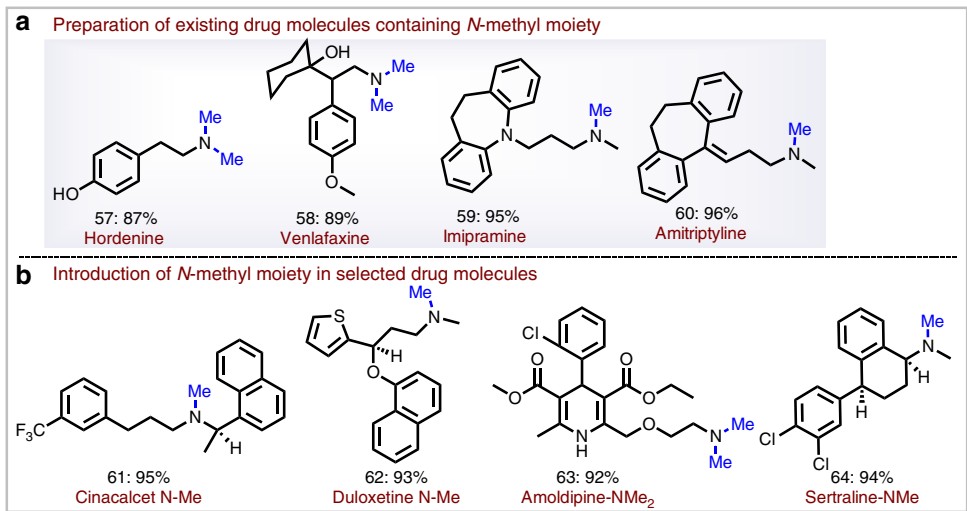

**Fig. 6** Synthesis of pharmaceutical *N, N*-(di)methylamines from the corresponding amines[i]. **a** [i]Reductive amination protocol demonstrating for the preparation of existing drug molecules. **b** Selective introduction of *N*-methyl moiety in drug molecules [i]Reaction conditions: 0.5 mmol amine, 50 mg Fe$_2$O$_3$/NGr@C (5 mol% Fe), 5 mmol of paraformaldehyde (150 mg), 0.5 mmol Na$_2$CO$_3$, 2 mL DMSO-water (1:1), 130 °C, 24 h, isolated yields

**Fig. 7** Gram-scale reactions. Demonstrating the practical utility for the synthesis of selected *N*-methylamines in up to 10 g. Reaction conditions: 1–10 g nitroarenes; 5 mol% Fe$_2$O$_3$/NGr@C, 300 mg paraformaldehyde (10 mmol) and 1 mmol Na$_2$CO$_3$ for each 0.5 mmol nitroarenes; 40–200 mL DMSO-water (1:1); 130 °C; 30–40 h; isolated yields

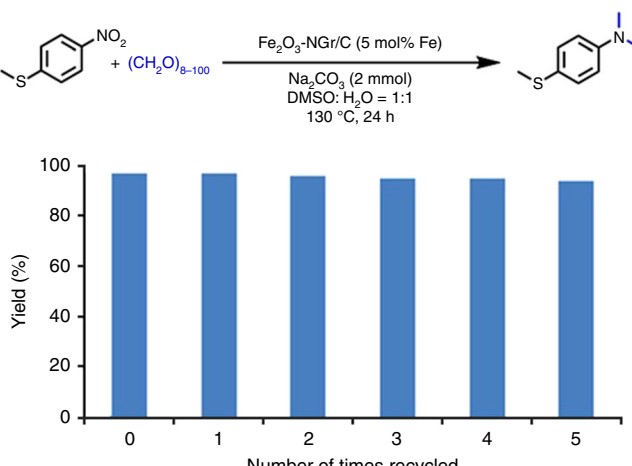

**Fig. 8** Catalyst recycling. Recycling and reusability of Fe$_2$O$_3$/NGr@C-catalyst for the *N, N*-dimethylation of 4-nitrothioanisole. [a]Reaction conditions: 1 mmol 4-Nitrothioanisole, 100 mg catalyst (5 mol%), 600 mg paraformaldehyde, 2 mmol Na$_2$CO$_3$, 4 mL DMSO-water (1:1), 130 °C, 24–30 h, yields were determined by GC

groups of the amine result selectively from paraformaldehyde and not from DMSO, which is used as reaction medium (Fig. 9b). Notably, in the overall reaction paraformaldehyde serves both as methylation and reducing agent for the one-pot reductive amination of nitroarenes (Fig. 9b, ii).

In conclusion, we have developed a straightforward, convenient and step economic reductive amino-methylation process starting from easily available nitroarenes using an earth-abundant and reusable iron oxide-based nanocatalyst. In this convenient pressure-free protocol, paraformaldehyde serves as both methylation and in situ generated hydrogen source that avoids the necessity of any specialized equipment and also use of additional reducing agents. Applying this operationally simple methodology, we synthesized a series of functionalized and structurally diverse *N*-methylamines. The synthetic utility of this reaction is specifically demonstrated using various life science molecules including existing pharmaceuticals.

## Methods

**General considerations.** Nitro compounds and amines were obtained from various chemical companies. Carbon powder, VULCAN® XC72R with Code XVC72R and CAS No. 1333-86-4 was obtained from Cabot Corporation Prod. TiO$_2$ (P25), Al$_2$O$_3$ and paraformaldehyde were obtained from Sigma-Aldrich. The pyrolysis experiments were carried out in Nytech-Qex oven.

**Fig. 9** Iron-catalyzed reductive *N*-methylation[a]. **a** Reaction pathway showing stepwise reductive methylation of nitro compounds. **b** [13]C-Labeled experiment showing paraformaldehyde as C1 source[i, ii, iii]. Reaction conditions: [i]0.5 mmol of 5-nitrobenzimidazole, 10 mmol of [13]C-labeled paraformaldehyde (300 mg), 50 mg $Fe_2O_3$/NGr@C (5 mol% Fe), 1 mmol $Na_2CO_3$ (106), 2 mL DMSO-water (1:1), 130 °C, 24 h. [ii]0.5 mmol of 1, 10 mmol paraformaldehyde (300 mg), 50 mg $Fe_2O_3$/NGr@C (5 mol% Fe), 1 mmol $Na_2CO_3$ (106 mg), 2 mL DMSO-water (1:1), 130 °C, 24 h. [iii]0.5 mmol of 1, 50 bar $H_2$, 50 mg $Fe_2O_3$/NGr@C (5 mol% Fe), 1 mmol $Na_2CO_3$ (106 mg), 2 mL DMSO-water (1:1), 130 °C, 24 h. All are isolated yields

TEM measurements were performed at 200 kV with an aberration-corrected JEM-ARM200F (JEOL, Corrector: CEOS). The microscope is equipped with a JED-2300 (JEOL) energy-dispersive x-ray-spectrometer (EDXS) for chemical analysis. The samples were deposited without any pre-treatment on a holey carbon supported Cu-grid (mesh 300) and transferred to the microscope. The High-Angle Annular Dark Field (HAADF) and Annular Bright Field (ABF) images were recorded with a spot size of approximately 0.1 nm, a probe current of 120 pA and a convergence angle of 30–36°. The collection semi-angles for HAADF and ABF were 70–170 mrad and 11–22 mrad, respectively.

XPS data were obtained with a VG ESCALAB220iXL (ThermoScientific) with monochromatic Al Kα (1486.6 eV) radiation. The electron binding energies $E_B$ were determined without charge compensation. For quantitative analysis the peaks were deconvoluted with Gaussian-Lorentzian curves, the peak areas were divided by a sensitivity factor obtained from the element specific Scofield factor and the transmission function of the spectrometer.

EPR spectra in X-band were recorded on a Bruker EMX CW-micro spectrometer equipped with an ER 4119HS-WI high-sensitivity cavity and a variable temperature control unit using the following parameters: microwave power = 6.64 mW, modulation frequency = 100 kHz, modulation amplitude = 1 G.

Mössbauer spectra were obtained at 300 K and 77 K by a Mössbauer spectrometer from Wissel GmbH equipped with a [57]Co source. Isomer shifts are given relative to α-Fe at room temperature. The spectra were analyzed by least-square fits using Lorentzian line shapes.

All catalytic experiments were carried out in ACE pressure (10 mL) tubes. GC conversions and yields were determined by GC-FID, HP6890 with FID detector, column HP530 m × 250 mm × 0.25 µm. NMR data were recorded on a Bruker ARX 300 and Bruker ARX 400 spectrometers.

**Preparation of $Fe_2O_3$/NGr@C-catalysts**. The $Fe_2O_3$/NGr@C-catalysts were prepared according to our previously reported procedures[19,44]. Appropriate amounts of $Fe(OAc)_2$ and 1,10-phenanthroline (phen; L1) corresponding to 3 wt% of Fe (1:3 molar ratio of Fe to phenanthroline) were stirred in ethanol for 30–40 min at room temperature. Then, carbon powder (VULCAN® XC72R) was added and the reaction mixture was stirred at 60 °C for 15 h. The reaction mixture was cooled to room temperature and ethanol was slowly removed in vacuo. The solid material obtained was dried at 60 °C for 12 h, after which was ground to a fine powder. Then, the grinded powder was pyrolyzed at the defined temperature (200, 400, 600, 800, or 1000 °C) for 2 h under an argon atmosphere and cooled to room temperature.

Elemental analysis of Fe-phenanthroline/C (Fe-phen/C-800, pyrolyzed at 800 °C for 2 h) (wt%): C = 91.1, H = 0.19, N = 2.69, Fe = 2.95.

The same procedure was applied for the preparation of Fe-with other nitrogen ligands such as 2,2′-bipyridine L2, 2,2′;6′,2″-terpyridine L3, and 2,6-bis(2-benzimidazolyl)pyridine L4 supported on carbon and also for L1 supported on $TiO_2$ and $Al_2O_3$.

The optimal catalyst used for the model studies and the preparation of the diverse amines has been previously characterized. For details see ref. [44] of this paper.

**Reductive methylation of nitroarenes using $Fe_2O_3$/NGr@C-catalysts**. An oven-dried 15 mL ACE pressure tube with stir bar was charged with $Fe_2O_3$/NGr@C (5 mol% Fe), paraformaldehyde (10–20 mmol), $Na_2CO_3$ (1 mmol), nitroarenes or amine (0.5 mmol), and dimethyl sulfoxide DMSO-water (1:1, 2 mL). The pressure tube was flushed with argon, closed with screw cap and the reaction was allowed to progress at 130–150 °C for desired time (24–40 h). After the completion of the reaction, the pressure tube was cooled down to room temperature and screw cap was opened. The catalyst was separated from the reaction products by filtration through celite. The separated catalyst along with celite was washed first with acetone and then with ethyl acetate. After solvent evaporation, the obtained crude product was purified by column chromatography (heptane: EtOAc) to afford the pure product which was submitted for analysis.

**[13]C- labeled experiment**. An oven-dried 10 mL pressure tube with stir bar was charged with $Fe_2O_3$/NGr@C (5 mol% Fe),[13]C-paraformaldehyde (10 mmol), $Na_2CO_3$ (1 mmol), 5-Nitrobenzimidazole (0.5 mmol), and dimethyl sulfoxide DMSO-water (1:1, 2 mL). The pressure tube was flushed with argon, closed with screw cap and the reaction was allowed to progress at 130 °C for 24 h. After the completion of the reaction, the pressure tube was cooled down to room temperature and screw cap was opened. The catalyst was separated from the reaction products by filtration through celite. The separated catalyst along with celite was washed first with acetone and then with ethyl acetate. After solvent evaporation, the obtained crude product was purified by column chromatography (heptane: EtOAc) to afford the pure product which was submitted for analysis

**Procedure for the gram scale reactions**. An oven-dried 100–1000-ml round bottom flask, equipped with stirring bar was charged with corresponding nitroarene, $Fe_2O_3$/NGr@C, paraformaldehyde, $Na_2CO_3$, DMSO-$H_2O$ (For reaction conditions see Fig. 8). The round bottom flask was flushed with argon and then fixed with reflux condenser. The reflux condenser was closed, flushed with Argon and the reaction was allowed to progress at 130 °C for desired time. After the completion of the reaction, the round bottom flask was cooled down to room temperature and reflux condenser was removed. The catalyst was separated from the reaction products by filtration through celite. The separated catalyst along with celite was washed first with acetone and then with ethyl acetate. After solvent evaporation, the obtained crude product was purified by column chromatography (heptane: EtOAc) to afford the pure product which was submitted for analysis.

**Procedure for catalyst recycling**. After the reaction similar to the procedure given in section S5.1, the catalyst was separated by centrifugation and thoroughly washed with distilled water, acetone and ethyl acetate sequentially in each run. Then, the washed catalyst was dried in high vacuum and was used for the next cycle. see Fig. 9 for the reaction conditions.

**Procedure for the detection of gases from paraformaldehyde**. $Fe_2O_3$/NGr@C (5 mol% Fe, 50 mg), paraformaldehyde (10 mmol), $Na_2CO_3$ (1 mmol), DMSO-water (1:1, 2 mL), and a magnetic stir bar were placed in a vial, which was then

capped with a septum equipped with a needle. The vial was placed in an autoclave, and heated at 130 °C for 4–5 h. Subsequently it was cooled down to room temperature and the reaction mixture was directly determined by GC gas phase analysis. See Supplementary Fig. 4 for the GC-spectrum for the detected gases:

**Data availability**. All data are available from the authors upon reasonable request.

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

## Acknowledgements

The Federal Ministry of Education and Research (BMBF) and the State of Mecklenburg-Vorpommern are gratefully acknowledged for their general support. We are grateful to Prof. A. Brückner, Drs. J. Radnik and M.-M. Pohl for catalyst characterization and thankful to the analytical staff of the Leibniz-Institute for Catalysis, Rostock for their excellent service.

## Author contributions

K.N., R.V.J., and M.B. planned and developed the project. K.N., R.V.J., and H.N. designed the experiments. K.N. performed all experiments, isolated and characterized the

products. R.V.J. and M.B. developed the catalysts. R.V.J., K.N., and M.B. wrote the paper. R.V.J. and M.B. supervised the project.

## Additional information

**Competing interests:** The authors declare no competing financial interests.

