## [Peer Review File · Nature Communications]

Reviewer #1 (Remarks to the Author):

The authors report the synthesis of advanced N-methylamines by reductive aminations. Cost-effective, stable, highly efficient, and reusable iron-based nano-catalysts were applied instead of noble metals. Stable and environmental-friendly paraformaldehyde instead of the widely used formaldehyde was used as a reagent which acts as both methylation and reducing agent (to form H₂ by decomposition over iron nanocatalyst), thus, in contrast to traditional methods using a two-step sequence, the production of N-methylamines was achieved with one-step process. This designed process does not necessitate the special equipment and the additional pressure of hydrogen. It also showed potential of scalable-production and easily controlled selectivity by the the concentration of paraformaldehyde and reaction time. This designed process could be extended to produce a broad range of substrates, including advanced life science intermediates, important drugs, fine chemicals, and so on. This strategy has many advantages compared with the traditional methods and represents a very important advance in the synthesis of N-methylamines. Therefore, I recommend the manuscript for publication in this journal.

Other comments:

1. Can the authors compare the activity of the two catalysts (Pd/C and Fe₂O₃/NGr@C) at room temperature?
2. The effect of the additional hydrogen pressure on the decomposition rate of paraformaldehyde was not analysed. The H₂ evolution rates from the decomposition of paraformaldehyde with- or without the addition of 4-nitroanisole should be investigated using Fe₂O₃/NGr@C catalyst since the authors claim that the small amounts of in situ generated hydrogen allow for selective reduction of nitroarene. Will the addition of 4-nitroanisole promote the decomposition of paraformaldehyde?
3. Is it possible to decide the rate-determining step for the reductive amination of 4-nitroanisole using paraformaldehyde as a methylation and reducing agent and Fe₂O₃/NGr@C as a catalyst without the additional hydrogen pressure?
4. The active site of the Fe₂O₃/NGr@C catalyst was not clearly revealed.
5. The n value of paraformaldehyde (CH₂O)_n should be given.
6. In Figure 3A, the text "F2O3" is wrong.
7. In Figure 4A, the text "X=H, heteroatom" is wrong. Should it read "X=C, heteroatom"?
8. Line 80: "Table 1, entries 9-10". It seems that there are no entries 9-10.
9. Figure 8, caption: the text "2 mL" should be removed from "40-200 mL DMSO-water (1:1, 2 mL)"?

Reviewer #2 (Remarks to the Author):

This is beautifully simple, benign, practical chemistry with impressive scope that would be of interest to scientists. The mysterious catalyst produced from relatively inexpensive materials (iron, graphite, phenanthroline) has impressive activity rivalling that of palladium and allowing a novel reaction. The catalyst enables the use of paraformaldehyde as a benign source of hydrogen (and carbon monoxide), the hydrogen being used in this case to reduce imines to amines. The chemistry allows the conversion of organic compounds already containing nitro groups into derivatives containing dimethylamino groups including those of interest to the pharmaceutical industry. The method introduced is somewhat less wasteful and benign than other methods, although a large of excess of paraformaldehyde is needed and one equivalent of carbonate base is consumed. The mystery surrounding the nature of the catalytic site that has the properties of palladium is a problem for this publication. No proposal of the source of the amazing catalytic activity is provided. In addition it is not clear why carbonate base is required.

There are a lot of mistakes in the presentation of the results which detract from the impact. Overall I have a mixed opinion

as whether to recommend publication in such a distinguished journal.

The discussion was somewhat confusing since it was not clear that the catalyst prepared at 800 degrees was the one that was used in all cases.

The method of isolation of the methylated products is not clear. I believe that it is the celite that is being washed after the reaction mixture has been filtered, not the reaction mixture itself. Is the organic product always dissolved in the DMSO/H₂O? Is the product extracted into acetone and ethylacetate? Which solvent is evaporated?

The authors should reference the supporting material from the 2013 Science paper (reference 43) instead of copying the same figures and discussion into the supporting material of the paper under review.

In each procedure in the supporting material, the amount of catalyst in mg should be included.

Corrections

Where does one obtain ACE pressure tubes?

material18-43 material18,43

Figure 2. works should be work (uncountable noun)

Figure 3. F₂O₃ should be Fe₂O₃ everywhere.

entries 9-10)! exclamation mark not needed.

Figure 9. Recycle time –recycle number is better (or Number of times recycled).

Reference 17 is incorrect.

In Figure 5, the molecules are inconsistently labelled; only Nicardipine-NMe₂ is labelled b.

Presumably the other compounds are reacted under condition a, but there is the need to clarify.

Similarly in Figure 6B, it is not clear what the condition d refers to (line 154). Should there be a change: the superscript b in line 159 to d? Where is d?

Provide references to noble metal catalyzed reductive aminations (line 47-48).

Typographical and English Errors

ethanol was slowly removed

room temperature and screw cap

The solid material obtained was dried at 60 °C for 12 hours, after which (it) was grinded (ground) to a fine

the grinded (ground) powder

- keep nomenclature consistent:

- sometimes N-methylamine has an unitalicized N (lines 32, 42, 208)

- sometimes methylamine is in two words as methyl amine (line 42)

- N,N-dimethylamine does not have a space after the comma (lines 105, 119, 123, 128, 134, 141, 143, 174, 183, 187, 198)

- in lines 125-126, the functional groups should be made plural, and OH and SH should be written out as alcohols and thiols for consistency

- change "screw" to "screw" (line 231)

- missing a period in line 234

supporting information:

define DBN, DIPEA

Table S4. Indicate that the catalyst prepared by heating Fe(phen) at 800 deg was used.

Procedure for the reparation (preparation) of

(same errors with grinded as above)

Reaction conditionsa:

General considerations

6 S1. General considerations

Appropriate amounts

was slowly removed

with lorentzian

KotBu DMSO

CS₂CO₃

scew cap was

619 HRMS (ESI): [M]⁺ (E.I., 70 eV) m/z (rel. int.) 281

Reply to the comments of the Reviewers

Reviewer #1 (Remarks to the Author):

The authors report the synthesis of advanced N-methylamines by reductive aminations. Cost-effective, stable, highly efficient, and reusable iron-based nano-catalysts were applied instead of noble metals. Stable and environmental-friendly paraformaldehyde instead of the widely used formaldehyde was used as a reagent which acts as both methylation and reducing agent (to form H₂ by decomposition over iron nanocatalyst), thus, in contrast to traditional methods using a two-step sequence, the production of N-methylamines was achieved with one-step process. This designed process does not necessitate the special equipment and the additional pressure of hydrogen. It also showed potential of scalable-production and easily controlled selectivity by the the concentration of paraformaldehyde and reaction time. This designed process could be extended to produce a broad range of substrates, including advanced life science intermediates, important drugs, fine chemicals, and so on. This strategy has many advantages compared with the traditional methods and represents a very important advance in the synthesis of N-methylamines. Therefore, I recommend the manuscript for publication in this journal.

Reply: We are thankful to this reviewer for recommending our paper for publication.

Other comments:

1. Can the authors compare the activity of the two catalysts (Pd/C and Fe₂O₃/NGr@C) at room temperature?

Reply: We performed the model amination reaction at room temperature using Pd/C and Fe₂O₃/NGr@C catalysts. In both cases there was no reaction observed and the starting substrate remained.

2. The effect of the additional hydrogen pressure on the decomposition rate of paraformaldehyde was not analysed. The H₂ evolution rates from the decomposition of paraformaldehyde with- or without the addition of 4-nitroanisole should be investigated using Fe₂O₃/NGr@C catalyst since the authors claim that the small amounts of in situ generated hydrogen allow for selective reduction of nitroarene. Will the addition of 4-nitroanisole promote the decomposition of paraformaldehyde?

Reply: We thank the reviewer for this suggestion. Obviously, decomposition of paraformaldehyde takes place in presence 4-nitroanisole. In order to investigate the decomposition of paraformaldehyde in absence of the aromatic substrate (4-nitroanisole), we performed experiments with paraformaldehyde in presence (GC spectrum 1) and absence of H₂ (GC spectrum 2) using our Fe₂O₃/NGr@C at 130 °C for 4-5 h. We observed that in both cases paraformaldehyde decomposed

and produced H₂, CO₂ and traces of CO. The GC spectra confirming the decomposition of paraformaldehyde is shown below. Notably, the procedure for the decomposition experiments and also spectra 2 was already included in the SI of the original manuscript (*S5.3 of original SI and 4.3 of revised SI*).

Data File C:\HPCHEM\1\DATA\2017\1702\170303\SIG16510.D (1) Sample Name: RVJ516

```

=====
Injection Date   : 03/03/2017 15:51:13 PM
Sample Name     : RVJ516
Acq. Operator  : RVJ
Location       : Vial 1
Inj           : 1
Inj Volume    : Manually

Acq. Method    : C:\HPCHEM\1\METHODS\WASSERST.M
Last changed   : 05/12/2016 14:31:22 PM by zw
Analysis Method: C:\HPCHEM\1\METHODS\CAL1602.M
Last changed   : 03/03/2017 14:28:07 PM by AA
                (modified after loading)
=====
  
```

(CH₂O)_n + Fe₂O₃ / water C + 10 bar H₂

External Standard Report

```

Sorted By      : Signal
Calib. Data Modified : 03/03/2017 14:28:05 PM
Multiplier    : 1.0000
Dilution      : 1.0000
Sample Amount  : 1.00000 [vol%] (not used in calc.)
  
```

Signal 1: FID1 A,

Signal 2: TCD2 B,

RetTime [min]	Type	Area [25 uV*s]	Amt/Area	Amount [vol%]	Grp	Name
2.719	BB	1.10103e4	8.63712e-3	95.09717		H2
7.790						Ar
11.238	BP	240.01024	2.11657e-4	5.07997e-2		CO
19.424	BP	180.27962	2.10195e-4	3.78939e-2		CH4
25.736	BV	2.01816e4	1.94290e-4	3.92108		CO2

Totals : 99.10694

Results obtained with enhanced integrator!
1 Warnings or Errors :

Warning : Calibrated compound(s) not found

*** End of Report ***

2

Data File C:\HPCHEM\1\DATA\2015\1501\150408\SIG15516.D

Sample Name: AR03-FeBr-Ir3

1 µmol FeBr2, IrPS, bpy, 5h, R2

=====
 Injection Date : 10/04/2015 14:01:51 PM
 Sample Name : AR03-FeBr-Ir3 Location : Vial 1
 Acq. Operator : AR Inj : 1
 Inj Volume : Manually
 Acq. Method : C:\HPCHEM\1\METHODS\WASSERST.M
 Last changed : 15/08/2013 09:05:09 PM by AK
 Analysis Method : C:\HPCHEM\1\METHODS\CAL1404.M
 Last changed : 10/04/2015 15:41:53 PM by CC
 (modified after loading)

$Fe_2O_3(Na_2CO_3 + (CH_2O)_n$
 $+ Na_2CO_3$ 2 µL $H_2O \Rightarrow DMSO$
 (1:1)

=====
 External Standard Report
 =====

Sorted By : Retention Time
 Calib. Data Modified : 10/04/2015 15:41:53 PM
 Multiplier : 1.0000
 Dilution : 1.0000
 Sample Amount : 1.00000 [vol%] (not used in calc.)

Signal 1: FID1 A,
 Signal 2: TCD2 B,

RetTime [min]	Sig	Type	Area	Amt/Area	Amount [vol%]	Grp	Name
3.433	2	PB	1789.18909	1.35596e-2	24.26065		H2
9.924	2	VV	1.03703e5	2.24577e-4	23.28933		Ar
12.619	2	VB	866.54584	2.23092e-4	1.93319e-1		CO
19.900	2		-	-	-		CH4
26.136	2	MM	1.57902e5	1.97618e-4	31.20429		CO2

Totals : 78.94759

Results obtained with enhanced integrator!
 1 Warnings or Errors :

Warning : Calibrated compound(s) not found

3. Is it possible to decide the rate-determining step for the reductive amination of 4-nitroanisole using paraformaldehyde as a methylation and reducing agent and Fe₂O₃/NGr@C as a catalyst without the additional hydrogen pressure?

Reply: We thank the reviewer for this suggestion. As pointed out in the manuscript this catalytic reductive amination reaction involves several steps (dehydrogenation of formaldehyde, several elementary reactions for the reduction of the nitro group, imine formation and final hydrogenation of the imine). We tried to elucidate the rate determine step under standard conditions but failed. We are sorry for this.

4. The active site of the Fe₂O₃/NGr@C catalyst was not clearly revealed.

Reply: Based on the detailed spectroscopic characterization of the active catalyst and non-active species we believe that the active site of the catalyst consists mainly of nanoscale Fe/Fe₂O₃ particles, which are surrounded by nitrogen species of the graphene layers. We think both the nano-sized particles and the Fe-N interactions are responsible for the remarkable activity of this catalyst.

5. The n value of paraformaldehyde (CH₂O)_n should be given.

Reply: We thank the referee for this suggestion. In the revised manuscript we have given the value of 'n' in the paraformaldehyde and it is 8-100: (CH₂O)₈₋₁₀₀

6. In Figure 3A, the text 'Fe₂O₃' is wrong.

Reply: We thank the reviewer for pointing out this mistake. This mistake has been corrected in the revised manuscript.

7. In Figure 4A, the text 'X=H, heteroatom' is wrong. Should it read 'X=C, heteroatom'?

Reply: We apologize for this mistake, which is corrected in the revised manuscript as; x= C or heteroatom.

8. Line 80: 'Table 1, entries 9-10'. It seems that there are no entries 9-10.

Reply: We apologize for this mistake. In the revised manuscript these entries have been removed.

9. Figure 8, caption: the text '2 mL' should be removed from '40-200 mL DMSO-water (1:1, 2 mL)'?

Reply: This mistake is removed in the revised manuscript.

Reviewer #2 (Remarks to the Author):

This is beautifully simple, benign, practical chemistry with impressive scope that would be of interest to scientists. The mysterious catalyst produced from relatively inexpensive materials (iron, graphite, phenanthroline) has impressive activity rivalling that of palladium and allowing a novel reaction. The catalyst enables the use of paraformaldehyde as a benign source of hydrogen (and carbon monoxide), the hydrogen being used in this case to reduce imines to amines. The chemistry allows the conversion of organic compounds already containing nitro groups into derivatives containing dimethylamino groups including those of interest to the pharmaceutical industry. The method introduced is somewhat less wasteful and benign than other methods, although a large excess of paraformaldehyde is needed and one equivalent of carbonate base is consumed. The mystery surrounding the nature of the catalytic site that has the properties of palladium is a problem for this publication.

No proposal of the source of the amazing catalytic activity is provided. In addition it is not clear why carbonate base is required.

Reply: We thank the reviewer for his/her comments on our work. The $\text{Fe}_2\text{O}_3/\text{NGr}@C$ catalyst is more reactive than commercial Pd/C. The active sites of this Fe-catalyst are found to be the nanoscale Fe_2O_2 particles, which are surrounded by nitrogen species of the graphene layers. Both the nano-size of the iron oxide particles and the Fe-N interaction are responsible for the activity of this material. However, in case of Pd/C catalyst there is no such Pd-N interactions and also the question of Pd-particles. Therefore, we believe the nano-size of iron oxide particles and also Fe-N interactions are responsible for this catalyst to exhibit more activity. The carbonate base is required for the decomposition of paraformaldehyde and also to abstract the proton from amine along with the imine hydrogenation.

There are a lot of mistakes in the presentation of the results which detract from the impact. Overall I have a mixed opinion as whether to recommend publication in such a distinguished journal.

Reply: We apologize for these formal mistakes and thank the reviewer for identifying several technical mistakes in the manuscript. In the revised manuscript we have tried to correct all these mistakes

The discussion was somewhat confusing since it was not clear that the catalyst prepared at 800 degrees was the one that was used in all cases.

Reply: For all successful reactions the catalyst pyrolyzed at 800 °C was used. This material gave the most active catalyst.

The method of isolation of the methylated products is not clear. I believe that it is the celite that is being washed after the reaction mixture has been filtered, not the reaction mixture itself. Is the organic product always dissolved in the DMSO/H₂O? Is the product extracted into acetone and ethylacetate? Which solvent is evaporated?

Reply: We apologize for the discrepancy in the procedure of isolation. We have the manuscript accordingly. The following detailed procedure is included in the revised manuscript: The catalyst was separated from the reaction products by filtration through celite. The separated catalyst along with celite was washed first with acetone and then with ethyl acetate. After solvent evaporation, the obtained crude product was purified by column chromatography (heptane: EtOAc) to afford the pure product which was submitted for analysis.

The authors should reference the supporting material from the 2013 Science paper (reference 43) instead of copying the same figures and discussion into the supporting material of the paper under review.

Reply: We thank the reviewer for this suggestion. Accordingly, we removed all the previous details of the characterization and added the reference (also reference 44 of the main manuscript) in Section 2 of the revised SI.

In each procedure in the supporting material, the amount of catalyst in mg should be included.

Reply: As requested the amount of catalyst was added in the SI of the revised version.

Corrections

Where does one obtain ACE pressure tubes?

Reply: Ace pressure tubes were obtained from Sigma Aldrich.

material 18-43 material18,43

Reply: This has been corrected.

Figure 2. works should be work (uncountable noun)

Reply: This has been corrected.

Figure 3. F₂O₃ should be Fe₂O₃ everywhere.

Reply: This has been corrected.

entries 9-10)! exclamation mark not needed.

Reply: Since there are no entries 9-10 in the revised manuscript, this sentence has been removed.

Figure 9. Recycle time –recycle number is better (or Number of times recycled).

Reply: In the revised manuscript 'Number of times recycled' has been given.

Reference 17 is incorrect.

Reply: We apologize for the mistake and this reference has been corrected in the revised manuscript.

In Figure 5, the molecules are inconsistently labelled; only Nicardipine-NMe₂ is labelled b. Presumably the other compounds are reacted under condition a, but there is the need to clarify. Similarly in Figure 6B, it is not clear what the condition d refers to (line 154). Should there be a change: the superscript b in line 159 to d? Where is d?

Reply: We thank the reviewer for pointing out this mistake. It has been corrected in the revised manuscript.

Provide references to noble metal catalyzed reductive aminations (line 47-48).

Reply: We provided a general reference for the reductive amination using noble/non-noble metal based catalysts. The reference no 18 in the revised manuscript has been included (Alinezhad, H., Yavari, H. and Salehian, F. Recent advances in reductive amination catalysis and its applications. *Curr. Org. Chem.* 19, 1021-1049 (2015).

Typographical and English Errors

ethanol was slowly removed

room temperature and scew cap

Reply: These errors have been corrected in the revised manuscript.

The solid material obtained was dried at 60 °C for 12 hours, after which (it) was grinded (ground) to a fine

the grinded (ground) powder

Reply: This was corrected in the revised manuscript.

- keep nomenclature consistent:

- sometimes N-methylamine has an unitalicized N (lines 32, 42, 208)

- sometimes methylamine is in two words as methyl amine (line 42)

Reply: These discrepancies have been corrected in the revised manuscript

- N,N-dimethylamine does not have a space after the comma (lines 105, 119, 123, 128, 134, 141, 143, 174, 183, 187, 198)

Reply: This was corrected in the revised manuscript.

- in lines 125-126, the functional groups should be made plural, and OH and SH should be written out as alcohols and thiols for consistency

Reply: This was corrected in the revised manuscript.

- change "scew" to "screw" (line 231)

- missing a period in line 234

Reply: This was corrected in the revised manuscript.

supporting information:

define DBN, DIPEA

Reply: DBN, DIPEA and others are defined in the revised SI. TEA = triethylamine, TMEDA = tetramethylethylenediamine, DBU = 1,8-diazabicyclo[5.4.0]undec-7-ene, DBN: 1,5-diazabicyclo[4.3.0]non-5-ene, DABCO = 1,4-diazabicyclo[2.2.2]octane, DIPEA = N,N-diisopropylethylamine.

Table S4. Indicate that the catalyst prepared by heating Fe(phen) at 800 deg was used.

Procedure for the reparation (preparation) of

(same errors with grinded as above)

Reply: Fe-Phen@C-800 was included in the revised Table S2 (previously it was table S4)

Reaction conditions a:

General considearations

6 S1. General considearations

Aproprate amounts

was slowly removed

with lorentzian

KotBu DMSO

CS₂CO₃

scew cap was

Reply: We are thankful to the reviewer for identifying these spelling mistakes. In the revised manuscript we corrected all these mistakes.

619 HRMS (ESI): [M]⁺ (E.I., 70 eV) m/z (rel. int.) 281

Reply: This was corrected in the revised manuscript.

Reviewer #1 (Remarks to the Author):

In the revised manuscript, the authors have basically addressed these issues raised by the referees. Therefore, I recommend acceptance of the manuscript.

Reviewer #2 (Remarks to the Author):

The authors have addressed my concerns. I recommend publication.